# Dynamical Behavior of Two Interacting Double Quantum Dots in 2D Materials for Feasibility of Controlled-NOT Operation

**DOI:** 10.3390/nano12203599

**Published:** 2022-10-13

**Authors:** Aniwat Kesorn, Rutchapon Hunkao, Kritsanu Tivakornsasithorn, Asawin Sinsarp, Worasak Sukkabot, Sujin Suwanna

**Affiliations:** 1Optical and Quantum Physics Laboratory, Department of Physics, Faculty of Science, Mahidol University, Bangkok 10400, Thailand; 2Department of Physics, Faculty of Science, Ubon Ratchathani University, Ubon Ratchathani 34190, Thailand

**Keywords:** double quantum dots, matrix model, electronic potential model, CNOT operation

## Abstract

Two interacting double quantum dots (DQDs) can be suitable candidates for operation in the applications of quantum information processing and computation. In this work, DQDs are modeled by the heterostructure of two-dimensional (2D) MoS_2_ having 1T-phase embedded in 2H-phase with the aim to investigate the feasibility of controlled-NOT (CNOT) gate operation with the Coulomb interaction. The Hamiltonian of the system is constructed by two models, namely the 2D electronic potential model and the 4×4 matrix model whose matrix elements are computed from the approximated two-level systems interaction. The dynamics of states are carried out by the Crank–Nicolson method in the potential model and by the fourth order Runge–Kutta method in the matrix model. Model parameters are analyzed to optimize the CNOT operation feasibility and fidelity, and investigate the behaviors of DQDs in different regimes. Results from both models are in excellent agreement, indicating that the constructed matrix model can be used to simulate dynamical behaviors of two interacting DQDs with lower computational resources. For CNOT operation, the two DQD systems with the Coulomb interaction are feasible, though optimization of engineering parameters is needed to achieve optimal fidelity.

## 1. Introduction

Realizing controllable quantum systems is of great interest and importance as their behaviors can unlock key understanding of other less controllable quantum systems, and lay the foundation for quantum technology applications. Physical systems, such as atoms, ions, spins, photons, and superconducting circuits, are used in quantum information processing, quantum simulation, and quantum computing [1,2,3]. Quantum dots (QDs) have been realized by many experimental approaches [4,5,6,7,8,9,10,11,12,13] and proven to possess advantages in terms of individual control, readout, and tunability [1] for quantum sensing, computing, and matter–light coupling in quantum communication [14,15].

The system of double quantum dots (DQD) consists of two quantum dots, each of which can constitute a two-level system known as a charge qubit, and in principle allows for a gate control desirable in quantum computing. Thus, the DQD behaviors, such as the electronic structure, and optical and dynamical properties, have been extensively and intensively investigated in theory, simulation and experiments. For example, in experiments, the charge qubit in DQD under high-speed rectangular voltage pulses was manipulated and the decoherence times were measured [4,5,6,7,8,9,10]. The electron transport in DQD was characterized [7,16]. The controlled-NOT (CNOT) operation of two strongly coupled semi-conductor charge qubits in GaAs/AlGaAs DQDs was demonstrated [11]. In simulation and theoretical studies, several methods, such as density functional theory (DFT), finite difference (FD), and tight binding (TB) methods, have been employed to investigate the behaviors of DQDs in various settings. DFT has been successfully adopted to study many quantum properties of 2D materials which are mostly obtained from bulk 3D materials. In particular, it has been used to calculate the band offsets in heterostructures of two-dimensional semiconductors [17], to calculate the energy band gap of 2D materials by various exchange correlation functionals to match experimental results reported [18], to adjust the band gap of the 2D transition metal dichalcogenides (2D-TMDs) by external strain [19,20], and to extract the electronic potential and effective mass of MoS_2_ [21], which can determine essential parameters of DQD models. Recently, DFT was deployed to investigate the stability of 18 monolayer metal oxides [22], from which 9 monolayer structures were predicted for the first time, and at least, 2D InO has been synthesized experimentally [23]. More relevantly, DFT and phonon calculations has enabled the investigation of electrical properties and dynamical stability of 2D materials XBi and XBi_3_ (X = B, Al, Ga, and In) [24], where the results suggested potentially compatible heterostructure systems. These systems exemplify heterostructures-based model systems with potential and viability for the purposes of the present work. Using the finite difference method, the effect of discontinuous effective mass was investigated in InAs/GaAs quantum dots [25], and, likewise, the energies of 2D-MoS_2_ periodic QDs [21]. Using the tight-binding method, the dynamics of the one charge qubit in InAs/GaAs DQD under external field was investigated [26,27].

Two-dimensional materials, e.g., MoS_2_, have exhibited great potentials in a wide range of applications, such as in optoelectronics [28,29], solar cells [30], batteries [31], and recently in quantum computation and information [1,32], single photon sources [19,33,34], sensing [35,36,37,38], etc. In particular, the QD structure has been constructed in 2D materials, and yielded great benefits in these applications [13] as well. To combine the advantages of QDs, for example, the fabrication of the heterostructure QDs in MoS_2_ have been demonstrated [21,39], where the 2H-phase MoS_2_ is changed to the 1T-phase with triangular shape when irradiated by an electron beam. The energy gaps calculated by the finite difference method showed good agreement with those from the experiment [21]. This paves the way for exploring quantum sensing and information processing with 2D materials both experimentally and computationally. For instance, using tight-binding and configuration interaction methods, the two-qubit system is generated from valley isospins of two electrons localized in the double quantum dot created within a MoS_2_ monolayer flake [32].

One of the essential processes in quantum information processing, especially quantum computing, is the evolution by gate operations from the initial state to the final state. Unitary single-qubit gates together with two-qubit CNOT gate form a universal set of gates, which enables any arbitrary computation, where other multiple-qubit gates can be decomposed into gates in this universal set [40]. CNOT gate has become a crucial gate in quantum computing. A physical system feasible for quantum computing must demonstrate its capability to create and execute the CNOT gate. From reports in literature, many two-level systems, such as superconducting qubits [41,42], photonic qubits [43,44,45,46,47,48,49], and charge qubits in DQDs [11,12] have shown CNOT gate control. The CNOT gate fidelity obtained is as high as 94.6% in superconducting qubits [42] and 74–84% in photonic qubits [48,49]. For charge qubits in DQDs, the Coulomb interaction between two DQDs is designed for CNOT gate operation which the fidelity is about 68% in GaAs/AlGaAs DQDs [11] and 63% in Si/SiGe DQDs [12]. A key obstacle is the interaction between two interacting DQDs which need thorough investigation into its behaviors, which can be chaotic.

In this work, we simulated the dynamical behaviors of two interacting DQDs under the influence of external electric field. The DQDs are modelled as the heterostructure of the 2D-MoS_2_ of 1T-phase and 2H-phase. The Hamiltonian of the two interacting DQDs are modeled by two models, namely the Coulomb electronic potential model and the two-level matrix model, whose matrix elements are obtained from the averaged interaction (partial trace) of the subsystem. The eigenenergies of DQD are investigated under various structural parameters: the base length *b* of QD, the inter-QD distance *d* between QDs, and the potential of the heterostructure. The dynamics of state are numerical simulation under situation of the CNOT operation and applied for the two qubit states. We remark that heterostructure of MoS_2_ is only exemplary, and it can be modified to accommodate other materials by changing the potential parameter in the simulation.

The remaining of this article is organized as follows. Section 2 outlines the methodology describing the DQD structure and the Hamiltonian construction with the electronic potential in the potential model and the calculation of matrix elements in the matrix model. It also includes qubit operations. Results and discussion are presented in Section 3, where the DQD energies and CNOT operation efficiency are reported and discussed. Finally, conclusions and final remarks are in Section 4.

## 2. Model and Methodology

### 2.1. Structural Model

According to experiment and simulation in Ref. [21], MoS_2_ in the semi-conducting 2H-phase changes to the metallic 1T-phase when irradiated by an electron beam. The transformed 1T-phase has a triangular shape, whose size depends on the radius of the beam, embedded in the rectangular 2H-phase (see Figure 1). This structure is a guiding model for simulation in this work, in which we can also investigate other parameters (e.g., QD dimensions, band offset of the heterostructure) and resulted behaviors. In experiment by Ref. [21], this heterostructure was achieved at room temperature. The periodic monolayers of 1T-phase and 2H-phase of MoS_2_ are calculated by DFT to evaluate the effective masses and potential parameters. The temperature was set to be 300 K in these DFT calculations. These effective masses and band offset values were used to construct the heterostructures, where the results of calculated energy gap (i.e., the energy difference of electron and hole) were compared favorably with those from experiments. We used these same values of parameters to model DQDs in our simulation. However, in our simulation, the dynamics of states are carried out by the time-dependent Schrödinger equation without decoherence and energy dissipation, which corresponds to zero temperature behavior. As reported in experiment by an annealing process [50], the 1T-MoS_2_ thin film changes significantly to the 2H-MoS_2_ phase at temperature higher than 498 K. Thus, at room temperature or lower, the considered heterostructure of 1T-MoS_2_ and 2H-MoS_2_ should be thermal stable.

For the model setting, the system of the 2H-phase MoS_2_ is assumed to be a rectangle of size dimensions $L_{x}$ nm and $L_{y}$ nm. The DQD is constructed from QDs with base length *b* nm and height *h* nm in the triangular shape of the 1T-phase MoS_2_. The QDs are placed with the inter-QD distance of *d* nm symmetric about the center of the rectangular $L_{x}\times L_{y}$ supercell. This constructs one DQD. In this work, two identical DQDs are placed side by side along the *x*-axis with a width *a* nm. We define the occupancy of an electron in the left dot and the right dot of the left DQD as the states ${|0\rangle }_{l}$ and ${|1\rangle }_{l}$, respectively. Likewise, the states ${|0\rangle }_{r}$ and ${|1\rangle }_{r}$ define the occupancy in the right DQD, as identified by the bits 0 and 1 in Figure 1.

### 2.2. Electronic Potential Model

The Hamiltonians of the system of two interacting DQDs are modeled by two methods; namely, (i) the electronic potential model and (ii) the matrix model. The former is more physical, but the latter is more computationally effective. After calibrating both models we can use the matrix model for dynamical simulation. In the electronic potential model, the 1T-phase and 2H-phase MoS_2_ shown by different colors in Figure 1 are represented by the electronic potential $V_{in}$ inside the QDs and $V_{out}$ outside the QDs, as described in Equation (Equation 1). For MoS_2_, the electron effective masses and potential parameters are taken from Ref. [21], in which these values were extracted from DFT calculations. The electron effective masses are $m_{e,2H}^{*}=0.54m_{e}$ for the 2H-phase (outside QD) and $m_{e,1T}^{*}=0.29m_{e}$ for the 1T-phase (inside QD), where $m_{e}$ is the mass of a free electron. The potentials of electron are $V_{in}=0$ inside the wells and $V_{out}=0.915\;\mathrm{eV}$ outside the wells.
(1)$$V\left(x,y\right)=\left\{\begin{array}{c} V_{in}\;\;;\mathrm{inside}\;\mathrm{QD}\\ V_{out}\;\;;\mathrm{outside}\;\mathrm{QD} \end{array}\right.$$

The Dirichlet boundary condition is applied for each DQD. The Hamiltonians of the *i*th (i=l,r) DQD are given in Equations (Equation 2) and (Equation 3). In these equations, the first term is the kinetic energy; the second term is the background potential energy of DQDs; the third term is the external potential from the applied electric field pulse; and the last term is the Coulomb interaction of an electron with another DQD. We abbreviate the last term from each equation as $I_{l}\left(x_{l},y_{l}\right)$ and $I_{r}\left(x_{r},y_{r}\right)$, respectively.
(2)$$\begin{array}{ccc} H_{l}\left(x,y\right) & = & -\frac{\hbar^{2}}{2m^{*}}\left(\frac{\partial^{2}}{\partial x_{l}^{2}}+\frac{\partial^{2}}{\partial y_{l}^{2}}\right)+V\left(x_{l},y_{l}\right)+V_{EF}\left(x_{l},y_{l},t\right)\\ & & +\frac{e^{2}}{4\pi \epsilon_{0}}\int \int \frac{|\psi_{r}\left(x_{r},y_{r},t\right){|}^{2}}{|{\vec{r}}_{l}-{\vec{r}}_{r}|}dx_{r}dy_{r} \end{array}$$
(3)$$\begin{array}{ccc} H_{r}\left(x,y\right) & = & -\frac{\hbar^{2}}{2m^{*}}\left(\frac{\partial^{2}}{\partial x_{r}^{2}}+\frac{\partial^{2}}{\partial y_{r}^{2}}\right)+V\left(x_{r},y_{r}\right)+V_{EF}\left(x_{r},y_{r},t\right)\\ & & +\frac{e^{2}}{4\pi \epsilon_{0}}\int \int \frac{|\psi_{l}\left(x_{l},y_{l},t\right){|}^{2}}{|{\vec{r}}_{l}-{\vec{r}}_{r}|}dx_{l}dy_{l} \end{array}$$

The Coulomb interactions $I_{l}\left(x_{l},y_{l}\right)$ and $I_{r}\left(x_{r},y_{r}\right)$ in Equations (Equation 2) and (3) have expensive computational cost when performed numerical calculation in each time step for dynamical evolution of the states of the DQDs. Therefore, the Coulomb interactions are approximated by Equations (Equation 4) and (Equation 5), respectively. Let $\psi_{l}\left(x_{l},y_{l},t\right)$ and $\psi_{r}\left(x_{r},y_{r},t\right)$ respectively denote the wavefunctions of the left and right DQDs. The qubit states ${|0\rangle }_{i}$ and ${|1\rangle }_{i}$ are represented by $\varphi_{i,0}\left(x_{i},y_{i}\right)$ and $\varphi_{i,1}\left(x_{i},y_{i}\right)$ which can be constructed by a linear combination of bonding and anti-bonding eigenstates of the non-interacting DQD. The vectors ${\vec{R}}_{i,0}$ and ${\vec{R}}_{i,1}$ are assumed at the centroid of each QD for representing the positions of the qubit states ${|0\rangle }_{i}$ and ${|1\rangle }_{i}$ of the *i*th DQD (here, i∈{l,r} indicates for the left and right DQDs). For the dynamics of a quantum state, the finite difference, effective mass and Crank–Nicolson method [51] is used to solve the time-dependent Schrödinger equation of the electronic potential model, as described in Equations (Equation 2)–(Equation 5).
(4)$$\begin{array}{ccc} I_{l}\left(x_{l},y_{l}\right) & \approx & \frac{e^{2}}{4\pi \epsilon_{0}}\left(\frac{1}{|{\vec{r}}_{l}-{\vec{R}}_{r,0}|}\times {\left|\int \int \varphi_{r,0}^{*}\left(x_{r},y_{r}\right)\psi_{r}\left(x_{r},y_{r},t\right)dx_{r}dy_{r}\right|}^{2}\right.\\ & & \left.+\frac{1}{|{\vec{r}}_{l}-{\vec{R}}_{r,1}|}\times {\left|\int \int \varphi_{r,1}^{*}\left(x_{r},y_{r}\right)\psi_{r}\left(x_{r},y_{r},t\right)dx_{r}dy_{r}\right|}^{2}\right) \end{array}$$
(5)$$\begin{array}{ccc} I_{r}\left(x_{r},y_{r}\right) & \approx & \frac{e^{2}}{4\pi \epsilon_{0}}\left(\frac{1}{|{\vec{r}}_{r}-{\vec{R}}_{l,0}|}\times {\left|\int \int \varphi_{l,0}^{*}\left(x_{l},y_{l}\right)\psi_{l}\left(x_{l},y_{l},t\right)dx_{l}dy_{l}\right|}^{2}\right.\\ & & \left.+\frac{1}{|{\vec{r}}_{r}-{\vec{R}}_{l,1}|}\times {\left|\int \int \varphi_{l,1}^{*}\left(x_{l},y_{l}\right)\psi_{l}\left(x_{l},y_{l},t\right)dx_{l}dy_{l}\right|}^{2}\right) \end{array}$$

### 2.3. Matrix Model

The charge qubits represented by two DQDs with the Coulomb interaction of electrons between the DQDs can be modeled by a 4 × 4 matrix. The Hamiltonian matrix in Ref. [11] is modified to become Equation (Equation 6): (6)$$H_{2q}=\left(\begin{array}{cccc} -\frac{1}{2}\left(\varepsilon_{l}+\varepsilon_{r}\right)+J_{2} & -\frac{1}{2}\Delta_{r} & -\frac{1}{2}\Delta_{l} & 0\\ -\frac{1}{2}\Delta_{r} & -\frac{1}{2}\left(\varepsilon_{l}-\varepsilon_{r}\right)+J_{3} & 0 & -\frac{1}{2}\Delta_{l}\\ -\frac{1}{2}\Delta_{l} & 0 & -\frac{1}{2}\left(-\varepsilon_{l}+\varepsilon_{r}\right)+J_{1} & -\frac{1}{2}\Delta_{r}\\ 0 & -\frac{1}{2}\Delta_{l} & -\frac{1}{2}\Delta_{r} & -\frac{1}{2}\left(-\varepsilon_{l}-\varepsilon_{r}\right)+J_{2} \end{array}\right.$$

Here, the Hamiltonian matrix is written in the basis |00〉, |01〉, |10〉 and |11〉, where $\varepsilon_{l}$ (resp. $\varepsilon_{r}$) is the energy detuning; $\Delta_{l}$ (resp. $\Delta_{r}$) is twice the inter-QD tunneling rate for the left (resp. right) DQD. The parameter ε can be modulated by the external electric field, and Δ is obtained by the energy difference between bonding and anti-bonding states of the non-interacting DQD from the electronic potential model above. $\sigma_{x}$ and $\sigma_{z}$ are the Pauli *X* and *Z* matrices, respectively. We note that the parameters $J_{1}$, $J_{2}$, and $J_{3}$ are the inter-qubit coupling energies defined by the Coulomb interaction: $J_{1}=e^{2}/4\pi \varepsilon_{0}r_{1}$, $J_{2}=e^{2}/4\pi \varepsilon_{0}r_{2}$ and $J_{3}=e^{2}/4\pi \varepsilon_{0}r_{3}$. These correspond to the distance $r_{1}=L_{x}+a-d$, $r_{2}=L_{x}+a$, and $r_{3}=L_{x}+a+d$, as illustrated in Figure 1. Hence, $J_{1}$, $J_{2}$, and $J_{3}$ are related.

The matrix Hamiltonian $H_{2q}$ in Equation (Equation 6) can be extracted for the subsystems, whose Hamiltonians are given by Equations (Equation 7) and (Equation 8), respectively.
(7)$$\begin{array}{ccc} H_{l} & = & -\frac{1}{2}\left(\varepsilon_{l}\sigma_{z}+\Delta_{l}\sigma_{x}\right)+J_{2}\left({\langle R|0\rangle }_{r}\langle 0{|}_{r}R\rangle \right){|0\rangle }_{l}{\langle 0|}_{l}\\ & & +J_{3}\left({\langle R|1\rangle }_{r}\langle 1{|}_{r}R\rangle \right){|0\rangle }_{l}{\langle 0|}_{l}+J_{1}\left({\langle R|0\rangle }_{r}\langle 0{|}_{r}R\rangle \right){|1\rangle }_{l}{\langle 1|}_{l}\\ & & +J_{2}\left({\langle R|1\rangle }_{r}\langle 1{|}_{r}R\rangle \right){|1\rangle }_{l}{\langle 1|}_{l} \end{array}$$
(8)$$\begin{array}{ccc} H_{r} & = & -\frac{1}{2}\left(\varepsilon_{r}\sigma_{z}+\Delta_{r}\sigma_{x}\right)+J_{2}\left({\langle L|0\rangle }_{l}\langle 0{|}_{l}L\rangle \right){|0\rangle }_{r}{\langle 0|}_{r}\\ & & +J_{3}\left({\langle L|1\rangle }_{l}\langle 1{|}_{l}L\rangle \right){|0\rangle }_{r}{\langle 0|}_{r}+J_{1}\left({\langle L|0\rangle }_{l}\langle 0{|}_{l}L\rangle \right){|1\rangle }_{r}{\langle 1|}_{r}\\ & & +J_{2}\left({\langle L|1\rangle }_{l}\langle 1{|}_{l}L\rangle \right){|1\rangle }_{r}{\langle 1|}_{r} \end{array}$$

The solution of the time-dependent Schrödinger equation with the governing Hamiltonian $H_{2q}$ in Equation (Equation 6) can be written in the form
(9)$$|\Psi_{2q}\left(t\right)\rangle =\alpha \left(t\right)|00\rangle +\beta \left(t\right)|01\rangle +\gamma \left(t\right)|10\rangle +\delta \left(t\right)|11\rangle .$$

However, Equations (Equation 7) and (8) constitute effective Hamiltonians for each subsystem, whose solutions can be, respectively, expressed as $|L\rangle ={a\left(t\right)|0\rangle }_{l}+b\left(t\right){|1\rangle }_{l}$ and $|R\rangle ={c\left(t\right)|0\rangle }_{r}+d\left(t\right){|1\rangle }_{r}$. Then, the solution of the composite system can be approximated as a product state $|\Psi_{ps}\rangle =|L\rangle ⊗|R\rangle$. In some cases, such as an entangled state, the state cannot be written as the product of the subsystems; hence, the Equations (Equation 7) and (Equation 8) cannot be used. If the state can be written as a product of the subsystem, the solution of $H_{2q}$ in Equation (Equation 6) and that from the product solutions of $H_{l}$ in Equation (Equation 7) and $H_{r}$ in Equation (Equation 8) are the same (see Appendix A).

In this work, the matrix model in Equations (Equation 7) and (Equation 8) is used to compare with the electronic potential model in Equations (Equation 2) and (Equation 3). The fourth order Runge–Kutta method [52] is used for solving the time-dependent Schrödinger equation of the matrix model.

### 2.4. Dynamics of States

#### 2.4.1. CNOT Operation

Two DQDs with the Coulomb interaction of electrons are simulated for the feasibility of CNOT operation. Here, the right qubit (i.d., right DQD) is used to control the left qubit (i.d., left DQD). The left qubit is prepared in the initial state ${|0\rangle }_{l}$, while the right (control) qubit can be prepared in the initial state ${|0\rangle }_{r}$ or ${|1\rangle }_{r}$. We need the initial state |01〉 to flip to the state |11〉, and the initial state |00〉 does not change under the operation. The states are initialized by the applied external electric field, as the right (control) qubit is fixed in the initial state by the strongly electric field, but the left qubit is operated by a square electric field pulse (strongly electric field for the initial state and rapidly switching to zero electric field for operation time).

#### 2.4.2. Transition Probability

In the electronic potential model, the DQDs have several electron eigenstates. For one DQD without an external electric field, the two lowest electron eigenstates are bonding states $\phi_{i,b}\left(x_{i},y_{i}\right)$ and anti-bonding state $\phi_{i,ab}\left(x_{i},y_{i}\right)$, respectively. The system is assumed to be a two-level system representing a qubit, and the higher levels are considered environment which can induce quantum leakage [26]. Since the state is written with probability amplitudes in the position space, we define the qubit states ${|0\rangle }_{i}$ and ${|1\rangle }_{i}$ in the position space as $\varphi_{i,0}\left(x_{i},y_{i}\right)$ and $\varphi_{i,1}\left(x_{i},y_{i}\right)$, respectively (*i* denotes the left or right qubits). Hence, the qubit states can be constructed by a linear combination of bonding and anti-bonding eigenstates of the DQD.
(10)$$\begin{array}{ccc} \varphi_{i,0}\left(x_{i},y_{i}\right) & = & \frac{1}{\sqrt{2}}\phi_{i,b}\left(x_{i},y_{i}\right)+\frac{1}{\sqrt{2}}\phi_{i,ab}\left(x_{i},y_{i}\right) \end{array}$$
(11)$$\begin{array}{ccc} \varphi_{i,1}\left(x_{i},y_{i}\right) & = & \frac{1}{\sqrt{2}}\phi_{i,b}\left(x_{i},y_{i}\right)-\frac{1}{\sqrt{2}}\phi_{i,ab}\left(x_{i},y_{i}\right) \end{array}$$

The dynamics of states for the electronic potential Hamiltonian in Equations (Equation 2) and (Equation 3) assume the form $|\Psi_{ps}\rangle =|L\rangle ⊗|R\rangle =\psi_{l}\left(x_{l},y_{l},t\right)⊗\psi_{r}\left(x_{r},y_{r},t\right)$ with the aforementioned |L〉 and |R〉. Therefore, the probability amplitudes in the qubit states ${|0\rangle }_{i}$ and ${|1\rangle }_{i}$ are given by $\langle \varphi_{i,0}\left(x_{i},y_{i}\right)|\psi_{i}\left(x_{i},y_{i},t\right)\rangle$ and $\langle \varphi_{i,1}\left(x_{i},y_{i}\right)|\psi_{i}\left(x_{i},y_{i},t\right)\rangle$, respectively.

## 3. Results and Discussion

### 3.1. Parameter Optimization for Energy Tuning of DQD

In this section, the DQD is modeled by the 2D electronic potential, as mentioned in the previous section. The electron eigenstates are computed, but we are mainly interested in the first bonding and anti-bonding eigenstates by tuning parameters (i.e., *b*, *d*, *V*) in the model. The numerical value of potential and the effective mass of MoS_2_ are obtained from Ref. [21]. The QD base length *b* is varied from 1.0 nm to 2.2 nm, and the height at h=b/2. The DQD energies converge when the supercell lengths ($L_{x}$ and $L_{y}$) are sufficiently large, as shown in Appendix A. In this simulation, the supercell of lengths $L_{x}=9.0\;\mathrm{nm}$ and $L_{y}=4.5\;\mathrm{nm}$ are selected.

In Figure 2, the wavefunctions of DQD are shown for the bonding state $\phi_{b}\left(x,y\right)$ and anti-bonding state $\phi_{ab}\left(x,y\right)$ for the QD base length b=2.0 nm and the inter-QD distance d=3.0 nm, where $x\in \{x_{l},x_{r}\}$ and $y\in \{y_{l},y_{r}\}$ depending on whether is considered the left or right DQD. The energies of the bonding and anti-bonding states are shown in Figure 3 when the inter-QD distance *d* varies in the *x*-axis. In Figure 3 also, the solid and dash lines of the same color (or symbol) denote the bonding and anti-bonding energies, respectively, whereas different colors (and symbols) represent lengths of the QD base *b*. The electron energy gap Δ is defined as the energy difference between the bonding and anti-bonding eigenstates, as plotted in Figure 4. The DQD is still completely separated with least distance when d=b (the QDs bases are joined); at this point, the energy gap is maximum.

We define the potential parameter $V=V_{out}-V_{in}$, so that V=0.915 eV for MoS_2_. Then, *V* is varied to cover the range of 0.60–2.00 eV, because some 2D materials have the energy band gap around 0–2 eV. Such variation can account for similar materials other than MoS_2_. Moreover, the external strain can adjust the band gap by a few tenths of eV around the original value [17,18,19,20]. Then, the energy gap is analyzed as a function of *V* and other engineering parameters. Below, we define a fitting function for the energy gap as a function of *V*, *b*, and *d* in the form:(12)$$\Delta \left(V,b,d\right)=\Delta_{max}\left(V,b\right)\exp\left[-\alpha \left(V,b\right)\left(d-b\right)\right].$$

The motivation for fitting with the above equation stems from analyzing 1D double quantum wells with the WKB approximation [53,54], in which case the energy gap decays exponentially as a function of the inter barrier width. Here, the decay rate also depends on the parameters of a single quantum dot; see Appendix A. In our case, the contour plots of maximum energy gap $\Delta_{max}\left(V,b\right)$ and the exponential component α(V,b) are presented in Figure 5a,b, respectively. In Figure 6 for a fixed *V*, the exponent α depends linearly on *b*
(13)$$\alpha \left(V,b\right)=m_{1}\left(V\right)b+c_{1}\left(V\right),$$
where $m_{1}$ and $c_{1}$ are the functions of *V*. It turns out that the slope $m_{1}$ obeys $m_{1}\left(V\right)=0.852V+0.295$, as shown in Figure 7, while $c_{1}\left(V\right)$ shows non-monotonic dependence on *V*. For more information are given in Appendix A.

We found that the energy levels always decrease as *b* increases, but the maximum energy gap $\Delta_{max}$ is not monotonic in *b*. Additionally, the energy gap always decreases when the inter-QD distance *d* increases, and the decay rate depends more strongly on *V* or *b*. Since the energy gap is a parameter in the matrix model for the dynamics simulation, the results of the energy gap dependence on the structural parameters and potential will be used in the matrix model. We emphasize that the energy gap discussed above is the energy difference between the electronic states, not the energy difference of the electron and hole.

### 3.2. Dynamics of States on Bloch Sphere

The dynamics of states are simulated to examine the CNOT operation by both models mentioned earlier. The solutions are written in the form of the product of the subsystems: $|\Psi_{ps}\rangle =|L\rangle ⊗|R\rangle =\psi_{l}\left(x_{l},y_{l},t\right)⊗\psi_{r}\left(x_{r},y_{r},t\right)$. The right qubit |R〉 is fixed in the initial state (either ${|0\rangle }_{r}$ or ${|1\rangle }_{r}$) as a controlling qubit. The left qubit |L〉 is prepared in the initial state ${|0\rangle }_{l}$ and operated by the external electric field. As examples, Figure 8 depicts the dynamics of states in the electronic potential model via the probability as a function of time in the two-qubit states for the initial states |01〉 and |00〉, respectively. The ideal qubit state, as defined in the Section 2, is assumed to be in a linear combination of the two-level system of the bonding and anti-bonding eigenstates of the DQD. Then, the initial states |0〉 or |1〉 are prepared by applying an external electric field in the *x* axis. If the applied electric field strength is weak, the initial state is in a superposition of |0〉 and |1〉. If it is too strong, the initial state may exit the QD, or it still remains inside the QD but in a superposition of higher energy levels other than the desired bonding and anti-bonding states. Therefore, the electric field strength should be varied for suitable preparation of the initial state for each DQD, as can be seen in Appendix A.

Furthermore, the dynamics of the states |L〉 and |R〉 can be represented by trajectories in the Bloch sphere, whose coordinates are calculated from $x_{l/r}\left(t\right)=Tr\left(\rho_{l/r}\left(t\right)\sigma_{x}\right),$$y_{l/r}\left(t\right)=Tr\left(\rho_{l/r}\left(t\right)\sigma_{y}\right)$ and $z_{l/r}\left(t\right)=Tr\left(\rho_{l/r}\left(t\right)\sigma_{z}\right)$ [2]. Here, the density matrices are $\rho_{l}\left(t\right)=\left|L\rangle \langle L\right|$ and $\rho_{r}\left(t\right)=\left|R\rangle \langle R\right|$. In Figure 9, the dynamics of states are represented in the Bloch sphere where the red and blue colors indicate the trajectories of the initial states |01〉 and |00〉, respectively. We note that if there is no inter-qubit interaction, the state |L〉 will precess around the *x*-axis of the Bloch sphere because it is an exact solution of the Hamiltonian in Equations (Equation 7) and (Equation 8) with $J_{1}=J_{2}=J_{3}=0\;\mathrm{eV}$. The inter-qubit interaction makes the precession of |L〉 around some axis lying in the xz plane, but the radius of the precession depends on the axis and the initial state.

The matrix model can be utilized to simulate the same situation, with significantly shorter time than that of the electronic potential model (by a factor of $10^{-3}$ or better). The comparison of simulated results from the two models are demonstrated in Figure 10 and in Appendix A. Due to increased effectiveness and reduced computational time in comparison to the electronic potential model, the matrix model is used to investigate the dynamics of the two interacting DQDs, in particular the performance of CNOT gate operation. It is worth noting that the matrix model contains only variables of energy, which can be calculated with high precision with other methods other than the finite difference. If the energy parameters are accurately determined, the matrix model prediction will improve.

### 3.3. CNOT Gate Efficiency

The two DQDs with the Coulomb interaction of electrons are used to construct a CNOT gate of two charge qubits, where the right qubit (DQD) is used to control the left one. To have a successful CNOT operation, the initial state |01〉 has to flip to the state |11〉, and the initial state |00〉 remains unchanged under the operation. The results are shown in Figure 8. Apparently, the operation is not perfect as expected. We define the parameters $P_{+}$ and $P_{-}$ for the maximum change in flipping probability of the initial state |01〉 and that for the initial state |00〉, respectively. Ideally, $P_{+}$ should tend to 1 and $P_{-}$ to 0 for high efficiency CNOT operation. Hence, the efficiency of the CNOT operation is defined by $\Delta P=P_{+}-P_{-}$ where ΔP=1 for a perfect CNOT operation. So that ΔP has the value between 0 and 1.

The CNOT operation efficiency ΔP is studied by varying the parameters, such as the QD base length *b*, the inter-QD distance *d*, the potential *V*, and the inter-DQD distance *a* in both models. As we mention earlier, the effective mass and potential parameters of MoS_2_ in Ref. [21] are used, but can be changed to other artificial potential values *V* (see Appendix A). In Figure 11 and Figure 12, the inter-DQD distance *a* is varied in the *x*-axis; there the separate panels correspond to different values of the QD base length *b*, and the inter-QD distance *d* is varied with different symbols (colors).

The two models are consistent, especially when the inter-qubit interaction is weak, e.g., the inter-DQD distance *a* and the inter-QD distance *d* are large. Additionally, the agreement between the two models is further improved in the regime of higher potential *V* (see Appendix A). The quantity ΔP is sensitive to the DQD parameters, and it has a turning point of local maximum when the DQD parameters are varied. Each curve of Figure 11 and Figure 12 is enumerated corresponding to the energy gap Δ, which is calculated from aforementioned DQD parameters. For each curve, the peak of ΔP depends on the inter-DQD distance *a*, which, in turn, depends on a set of inter-qubit coupling energies {$J_{1}$, $J_{2}$, $J_{3}$}. Then, the maximum values of ΔP are extracted as a function of Δ, and their relationship is plotted in Figure 13. Thus, for a selected curve of ΔP, there is a set of {$J_{1},J_{2}$, $J_{3}$} which gives the maximum ΔP. As mentioned earlier, $J_{1}$, $J_{2}$, and $J_{3}$ are related, and $J_{2}$ lies between $J_{1}$ and $J_{3}$. We choose $J_{2}$ to represent the set of the inter-qubit coupling energies to investigate the peak of ΔP with respect to the strength of interaction. In principle, $J_{1}$ or $J_{3}$ could have been chosen, but it is more reasonable to represent by the parameter near the average, which also corresponds to the distance between the centers of two DQDs (although there is no charge at the centers). The parameter $J_{2}$ that yields the maximum ΔP are plotted as a function of the energy gap Δ in Figure 14. Both $J_{2}$ and Δ affect ΔP. At the maximum ΔP, $J_{2}$ and Δ exhibit a linear relationship. The high energy gap requires the high inter-qubit coupling energy, but the inter-qubit coupling energy tends to the maximum when the two DQDs join at the base of the supercell (a=0). After this point, $J_{2}$ cannot increase even as the energy gap increases, as shown by the saturation of curves in Figure 14. Consequently, the maximum ΔP decreases at the higher energy gap, as shown in Figure 13.

In the matrix model, ΔP is quite sensitive to the relation of the set of the inter-qubit coupling energy. Hypothetically, suppose we change the parameters $J_{1}$, $J_{2}$, and $J_{3}$ (with conforming to the aforementioned relationship), say, $J_{1}$ is increased by 5% to 15% while the others are kept unchanged, then ΔP can increase by almost 2 to 3 folds. In the matrix model with extreme case, such as small $J_{2}\approx J_{3}$ and very strong $J_{1}\gg \Delta$, the CNOT operation shows high efficiency with ΔP reaching nearly 1 (see Appendix A). However, in a realistic model, $J_{1}$, $J_{2}$, and $J_{3}$ cannot change independently. Moreover, the inter-qubit interaction in this simulation is not precisely determined since the other effects are not considered, such as the screening Coulomb interaction when the DQDs are placed in the permittivity dependent environment [55,56,57,58].

In Section 3.2, we have shown the electronic potential and matrix models yield consistent dynamics of states, and likewise the CNOT gate efficiency as indicated by ΔP. However in literature, the CNOT gate efficiency is often reported with the average fidelity $F_{av}$ comparing the ideal CNOT gate and experimental/simulation one. Let $M_{\mathrm{CNOT}}^{\mathrm{ideal}}$ and $M_{\mathrm{CNOT}}^{\mathrm{sim}}$, respectively, denote the matrices of the CNOT operators for the ideal and one constructed from simulation. According to Ref. [59], the average fidelity $F_{av}$ can be calculated from:(14)$$F_{av}=\frac{1}{n(n+1)}\left[Tr\left(MM^{†}\right)+{\left|Tr(M)\right|}^{2}\right],$$
where $M={\left(M_{\mathrm{CNOT}}^{\mathrm{ideal}}\right)}^{†}M_{\mathrm{CNOT}}^{\mathrm{sim}}$, and n=4 for the dimension of the Hilbert space for two qubits. In dynamical simulation by the matrix model, the operator $M_{\mathrm{CNOT}}^{\mathrm{sim}}$ can be constructed, and so the average fidelity $F_{av}$ as a function of the inter-DQD distance *a* can be computed along with ΔP, as shown in Figure 15. In such cases, the peaks of the average fidelity are attained around 54% to 57%. To put in perspectives, the obtained fidelity from this work is slightly lower than previously reported in DQD experiments [11,12]. However, it should be emphasized that such comparison is not meaningful since the materials and methodology are different, but it indicates that a sensible figure is obtained. We remark that ΔP in Figure 8 is the change of probability from the highest to the lowest, which may occur at different times depending on the initial state. For the constructed $M_{\mathrm{CNOT}}^{\mathrm{sim}}$, the operation time which yields the maximum flipping probability of the initial state |01〉 is also used at the operation time of other initial states. If ΔP is computed at a fixed operation time (e.g., the operation of $M_{\mathrm{CNOT}}^{\mathrm{sim}}$), then ΔP also shows similar behaviors (e.g., discontinuity) like the average fidelity, shown with dash lines in Figure 15. Thus, $F_{av}$ and ΔP both can indicate the efficiency of CNOT gate operation. We additionally remark that the discontinuity of the fidelity is sensitive to the inter-qubit interacting {$J_{1},J_{2}$, $J_{3}$}, in the sense that arbitrary increasing $J_{1}$ with a constant multiple, while keeping others parameters fixed, the discontinuity disappears.

In summary, the two interacting DQDs with the Coulomb interaction in the heterostructure of materials, such as MoS_2_ can be constructed and optimized for CNOT gate operation.

## 4. Conclusions

The DQDs are modeled by a heterostructure of two dimensional MoS_2_ consisting of the 1T-phase triangular shape embedded in the 2H-phase square supercell. The two interacting DQDs are investigated for the feasibility of CNOT gate operation.

The Hamiltonian of the system is modeled by the 2D electronic potential and 4×4 matrix models. The DQDs in the electronic potential model and spatial dependent effective mass are studied with finite difference for DQD energy tuning, where the energy difference between the bonding and anti-bonding electronic eigenstates is maximized as a function of the electronic potential *V* and the QD base length *b*. The energy gaps can be explained well with the WKB approximation for quantum double wells, showing exponential decay depending on the inter-QD distance *d*, which decreases more rapidly when *V* and *b* increase. This information can be used to examine other QDs with different size or material make-up.

The two DQDs with the inter-DQD Coulomb interaction of electrons can be used to construct two interacting charge qubits with possible CNOT gate operation. The performance of CNOT operation via the dynamics of the two charge qubits are simulated by the Crank–Nicolson method in the potential model and by the fourth order Runge–Kutta method in the matrix model. For the comparison of the computational techniques, the matrix model ensures lower computational cost than the potential model, thus leading to the faster calculation. The results of the two models are in excellent agreement, and both show low CNOT operation efficiency ΔP with the pure Coulomb interaction. When varying the DQD parameters, the CNOT operation efficiency ΔP exhibits a peak of local maximum, which suggests that the engineering parameters can be tuned to optimize it. Additionally, the CNOT operation efficiency is reported with the average fidelity $F_{av}$ which exhibits the same trend as ΔP.

Finally, we believe that two interacting double quantum dots can be viable candidate for CNOT gate operation after selecting optimized DQD parameters, and our work sheds some light on how the behaviors of two interacting DQDs for CNOT operation based on QD systems depend on these parameters. This computational study can be beneficial in designing experiments in DQDs.

## Figures and Tables

**Figure 1 nanomaterials-12-03599-f001:**
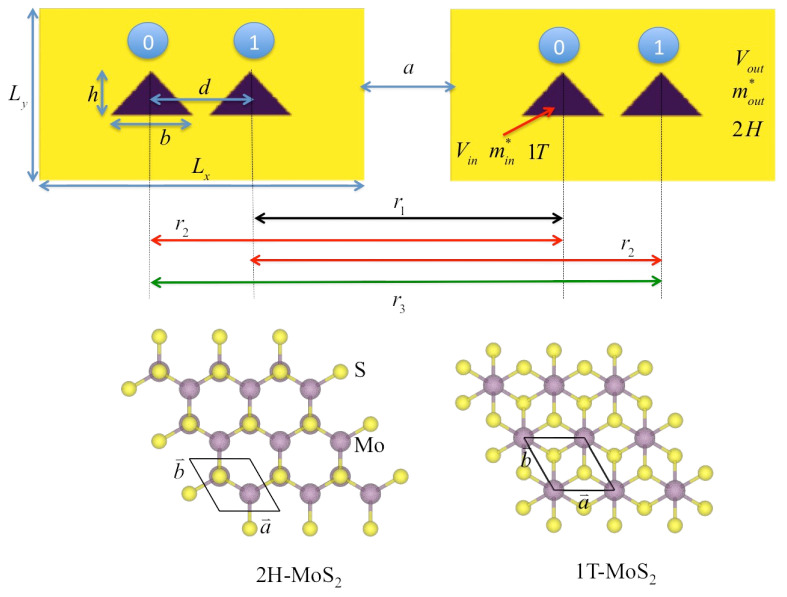
Schematic diagram of 2D two DQDs which different colors representing the different electronic potentials. This is an illustration of MoS_2_ in Ref. [21]. Yellow rectangles denote the 2H phase, and violet triangles denote the 1T phase.

**Figure 2 nanomaterials-12-03599-f002:**
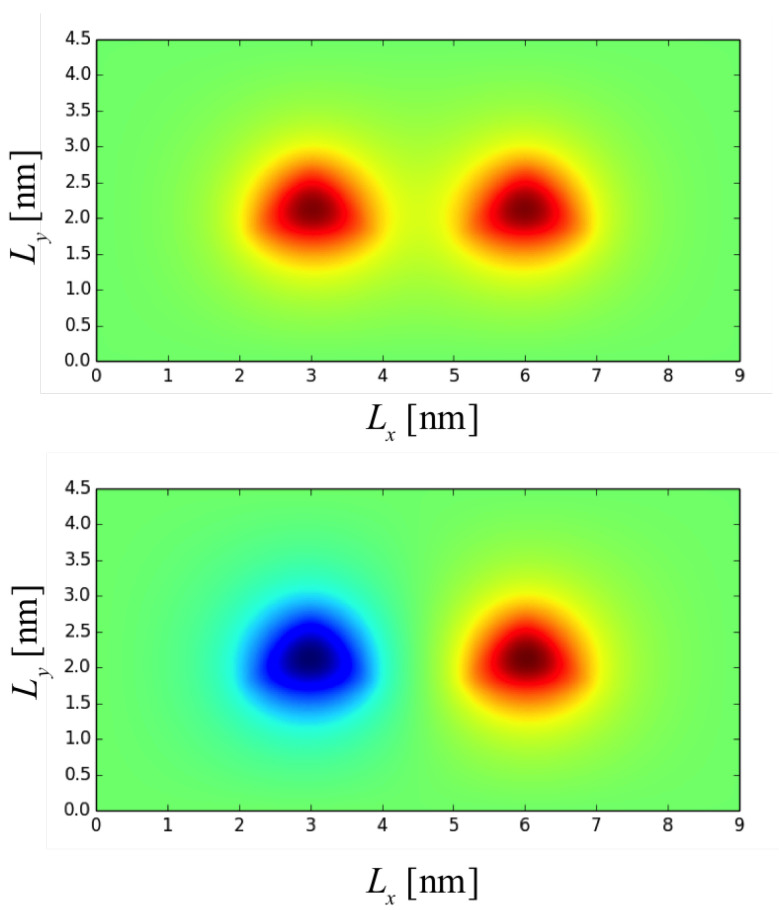
The wavefunction of MoS_2_ DQD with QD base length b=2.0nm and inter-QD distance d=3.0nm. (**top**) bonding state and (**bottom**) anti-bonding state.

**Figure 3 nanomaterials-12-03599-f003:**
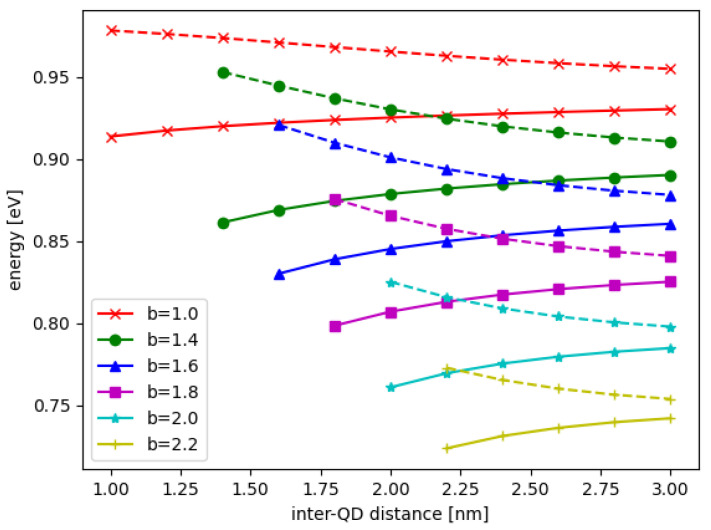
The DQD energies as a function of inter-QD distance *d*; the solid and dash lines represent the energies of the bonding and anti-bonding states, respectively. Different lengths of QD base *b* are shown in different colors and symbols.

**Figure 4 nanomaterials-12-03599-f004:**
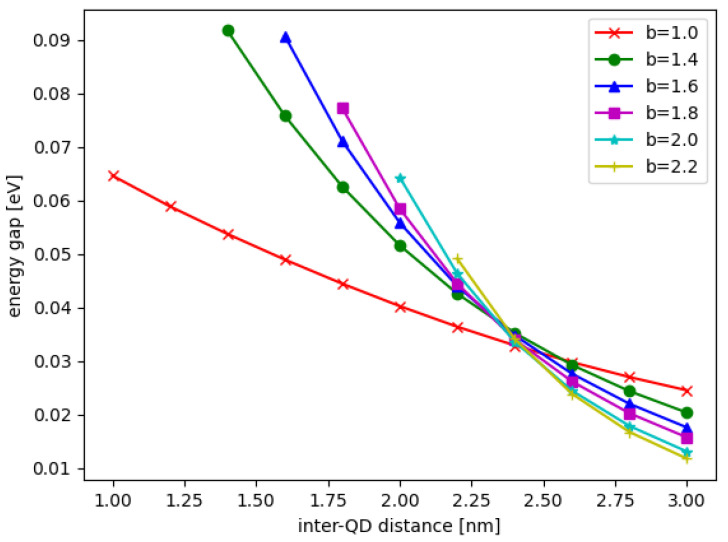
The energy gap Δ between the bonding and anti-bonding states of electron as a function of the inter-QD distance *d*, the different lengths of QD base *b* are shown in different colors and symbols.

**Figure 5 nanomaterials-12-03599-f005:**
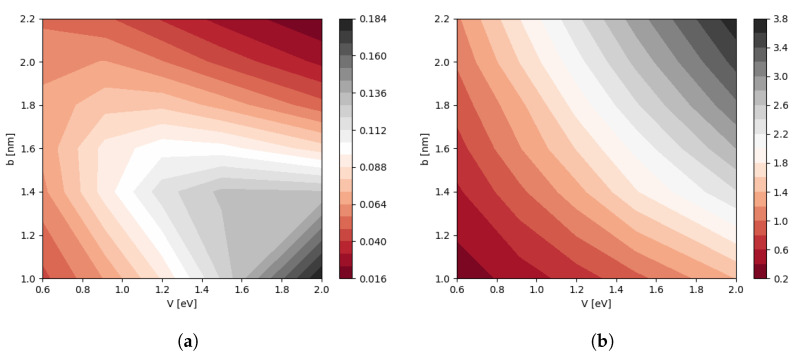
(**a**) The contour of maximum energy gap $\Delta_{max}$ which defined at d=b. (**b**) The contour of fitting exponential parameter α which modeled in Equation (Equation 12). The x-axes and y-axes are the potential *V* and QD base *b*, respectively.

**Figure 6 nanomaterials-12-03599-f006:**
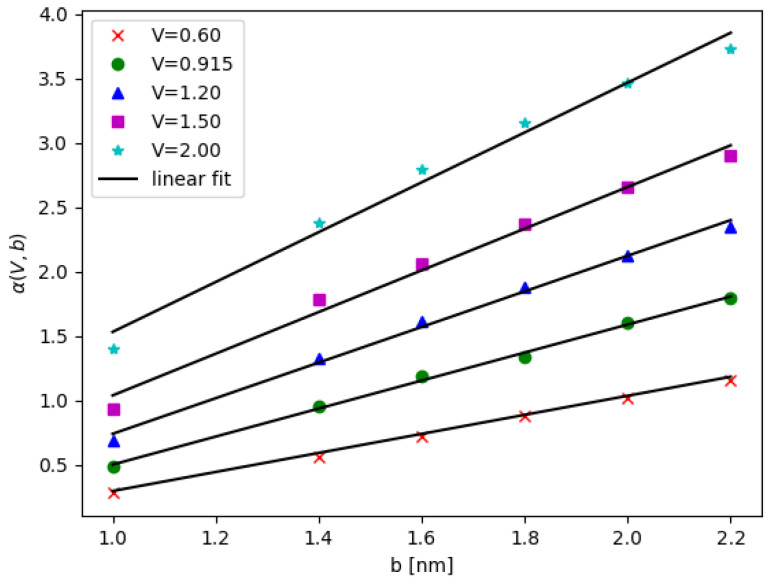
The component α as a function of QD base length *b* with the different values of potential *V*, the solid lines are the linear fitting.

**Figure 7 nanomaterials-12-03599-f007:**
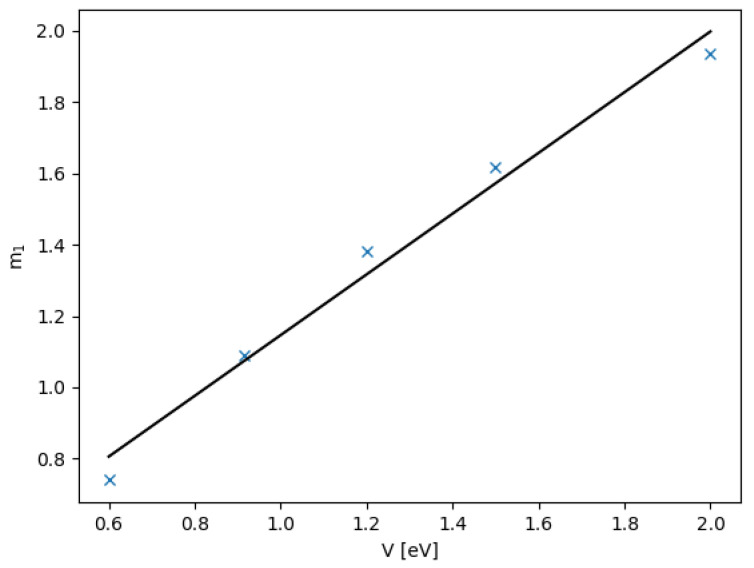
The slope $m_{1}$ as a function of the potential *V* with linear fitting in solid line as $m_{1}\left(V\right)=0.852V+0.295$.

**Figure 8 nanomaterials-12-03599-f008:**
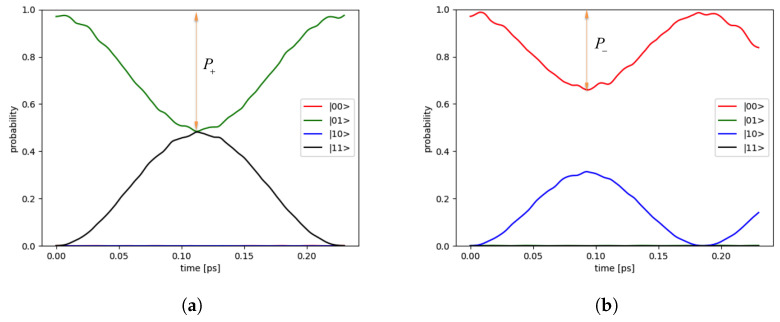
The dynamics of states are simulated for CNOT operation with the electronic potential model. The probability as a function of time in two qubit states are shown for (**a**) initial state |01〉 and (**b**) initial state |00〉.

**Figure 9 nanomaterials-12-03599-f009:**
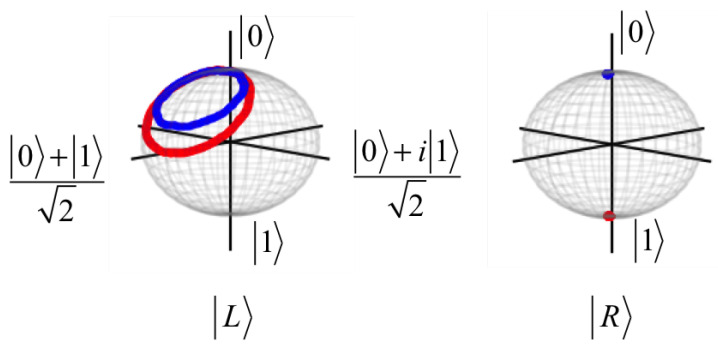
The dynamics of states |L〉 and |R〉 displayed in the Bloch sphere for the left and right qubits, corresponding to Figure 8. The solution of the composite system is written as a product state $|\Psi_{ps}\rangle =|L\rangle ⊗|R\rangle$. Red and blue colors are the trajectories of the initial states |01〉 and |00〉, respectively. The right qubit state |R〉 is fixed in either ${|0\rangle }_{r}$ or ${|1\rangle }_{r}$ as a control qubit.

**Figure 10 nanomaterials-12-03599-f010:**
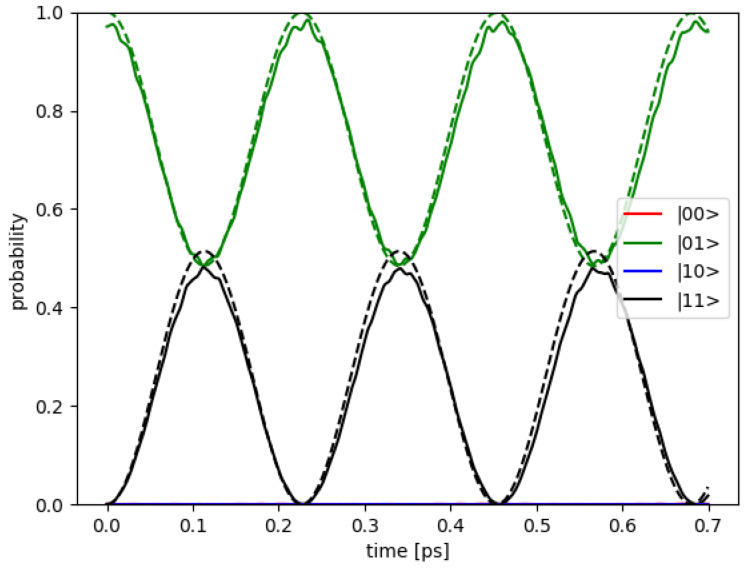
The dynamics of states of two DQDs with the inter-DQD Coulomb interaction for the initial state |01〉, the QD base length b=2.0nm, the inter-QD distance d=3.0nm and the two DQDs are separated with the inter-DQD distance a=8.0nm. The solid and dash lines are electronic potential and matrix models, respectively.

**Figure 11 nanomaterials-12-03599-f011:**
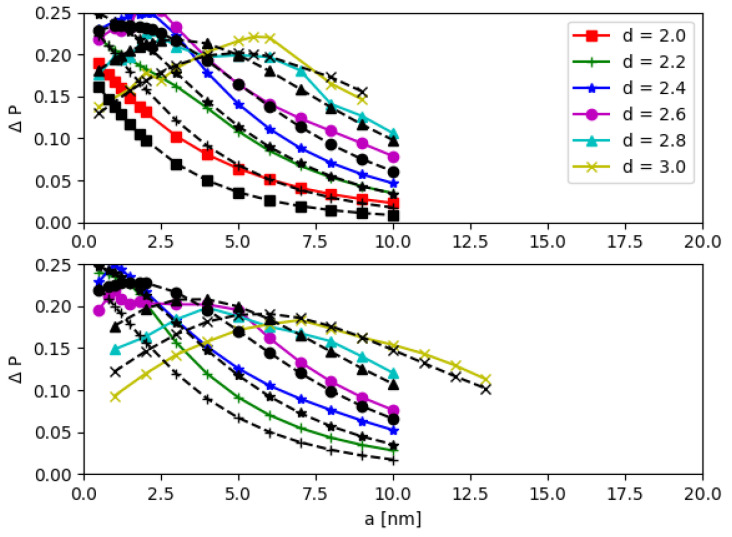
CNOT operation efficiency ΔP as a function of the inter-DQD distance *a*, the QD base length b=1.6nm and b=1.8nm for the top and bottom panels, respectively. The different inter-QD distance *d* is depicted by different symbols. The black dash lines and other color solid lines are from the matrix and electronic potential models, respectively.

**Figure 12 nanomaterials-12-03599-f012:**
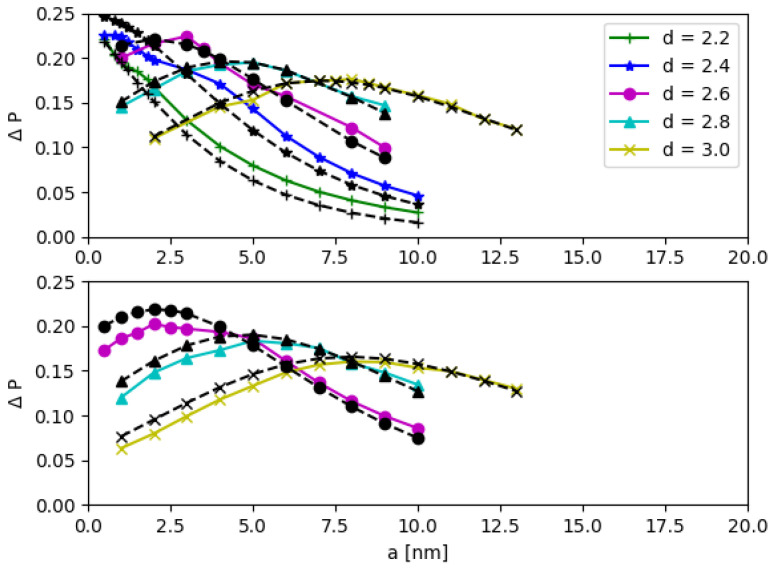
CNOT operation efficiency ΔP as a function of the inter-DQD distance *a*, the QD base length b=2.0nm and b=2.2nm for the top and bottom panels, respectively. The different inter-QD distance *d* is depicted by different symbols. The black dash lines and other color solid lines are from the matrix and electronic potential models, respectively.

**Figure 13 nanomaterials-12-03599-f013:**
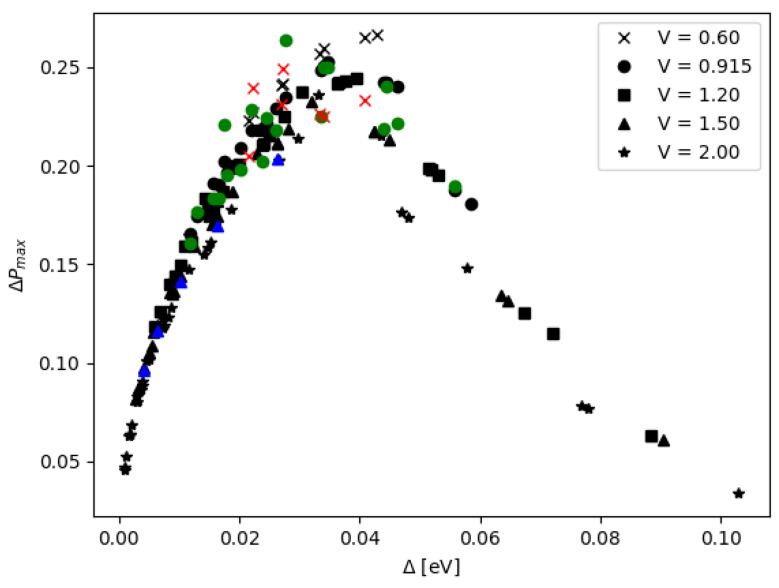
Maximum ΔP corresponding to the DQDs with the energy gap Δ; values from the matrix model are in black and those from the electronic potential model are in other colors.

**Figure 14 nanomaterials-12-03599-f014:**
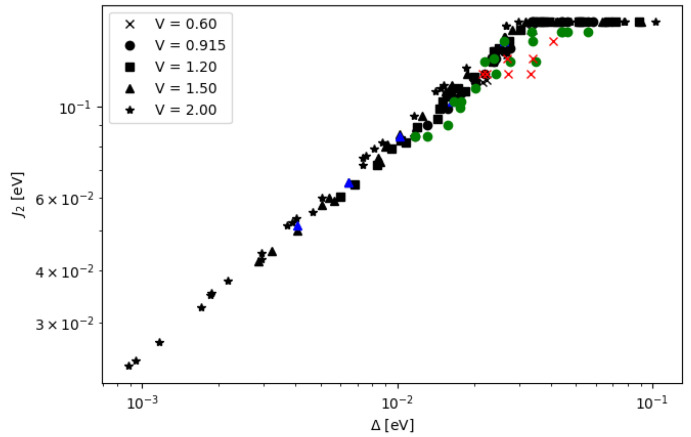
The inter-qubit coupling energy $J_{2}$ and the energy gap Δ which give the maximum ΔP of the DQDs are plotted in the log-log scale. Calculations from the matrix model are in black and those from the electronic potential model are in other colors.

**Figure 15 nanomaterials-12-03599-f015:**
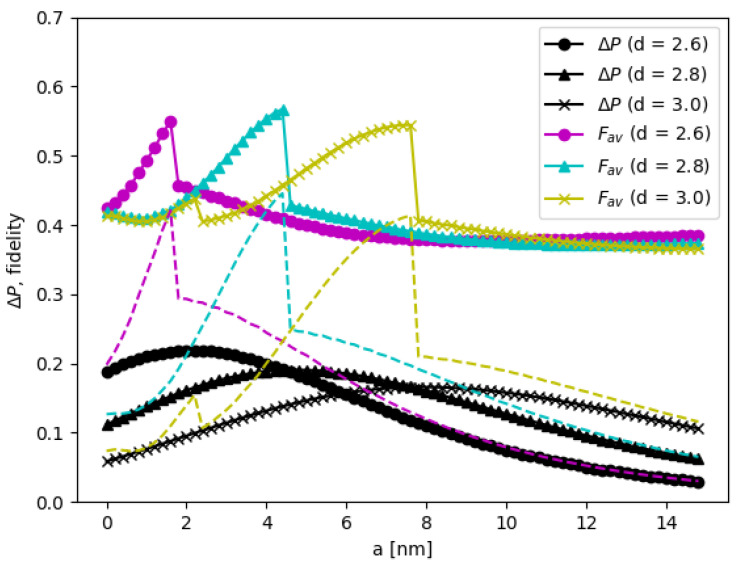
CNOT gate efficiency with the average fidelity $F_{av}$ and ΔP as a function of the inter-DQD distance *a*. The DQD potential is MoS_2_, and the QD base length b=2.2nm. The inter-QD distance *d* is depicted by different symbols.

## Data Availability

All data that support this study are included within the article and any Appendix A.

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
