# Peer review of "Dynamical Behavior of Two Interacting Double Quantum Dots in 2D Materials for Feasibility of Controlled-NOT Operation"

_nanomaterials, 2022, doi:10.3390/nano12203599_

Round 1
Reviewer 1 Report
The authors report on a simulation of the dynamical behaviour of two double quantum dots with a view of its applicability as a CNOT gate.
The research is timely and interesting. I believe that the work should be published in Nanomaterials (subject to the minor amendments listed below) in order that the theory can be tested against experiment.
(1) Line 26: "in principle"
(2) Line 29: "for example"
(3) Lines 30/31: "was manipulated"
(4) Lines 44/45: This sentence is incomplete. E.g. "... was investigated."
(5) Line 49: "has been"
(6) Between lines 107 and 108: "m_e is the mass of an electron" should probably read "m_e is the mass of a free electron"
(7) Between lines 107 and 108: For the values of the effective masses Ref. 21 is quoted, however where does the value of the employed band offset (0.915 eV) come from? And do these values take the reduced dimensionality of the system into account?
(8) Line 130: "basis"
(9) Equation (7): The two vertical lines should probably be two "+" symbols.
(10) Line 145: "cannot be used"
(11) Figure 2: I presume that the axis labels are in units of "nm". Perhaps this should be mentioned.
(12) Line 250: "tend to"
(13) Line 331: "lower"
Reviewer 2 Report
This manuscript is dedicated to demonstrating simulation results on the dynamical behavior of two double quantum dots (DQD) in 2D Materials featuring Coulomb Interaction. The DQD are modeled using heterostructure of two dimensional (2D) MoS2 which is an interesting approach from both theoretical and application point of view and has not been studied well yet. Aspects of the simulations are very well systematized. The discussion provided is quite adequate for the present purpose. The well detailed and at the same time comparative context of the simulation results clarifies convincingly the researched approach and prompts a good understanding of the dynamical behavior of the DQD which can benefit wide range of developing research and applications in the field of quantum information processing and computation.
From practical point of view, the reported results thus bring new knowledge and certainly represent an original contribution in the present context.
The authors chose an adequate structure of the manuscript – an excellent point of departure for such a study. Also, they provided a balanced realistic and nicely illustrated presentation of their results and corresponding analysis that is of much scientific and practical interest and adds new knowledge to the field.
The present manuscript is a significant contribution, this work once published would be instructive and suggestive in terms of further studies and to a wider readership.
There are some minor issues with this already excellent manuscript that will need to be addressed before becoming suitable for publication, i.e., it can be considered for publication after a minor revision:
1: Title is a little bit heavy and not attractive to wider audience, should be shortened and made more focused; The word “behavior” is usually used in singular. The abbreviation “CNOT” is not so appropriated for a title.
2: The “CNOT” abbreviation although well known in specialized circles, should be clearly defined in the abstract and in the main text separately for the benefit of wider readership audience.
3: In the introduction, the authors miss that DFT (and even higher) levels of theory have already been adopted/used for studying the quantum aspects of similar heterostructures as the one used and employed in the present work. Examples in which such theoretical works help understanding synergies and quantum issues and even directly guide experimental work include Journal of Physics: Condensed Matter 27 (2015) 485306 and Applied Surface Science 548 (2021) 149275. Such works should be referenced for achieving a clear picture of viability of heterostructures-based model systems for the purposes of the present work.
4: Authors should mention at what general temperatures conditions the simulations are intended for. Also, the authors should elaborate and be more specific about any concrete evaluation of the thermal stability range applicable to the employed heterostructures.
5: Spell-check and stylistic revision of the paper are still necessary. Some, long sentences, misspellings, etc., still are noticeable throughout the text.
